# The Long-Distance Transport of Jasmonates in Salt-Treated Pea Plants and Involvement of Lipid Transfer Proteins in the Process

**DOI:** 10.3390/ijms25137486

**Published:** 2024-07-08

**Authors:** Gulnara Vafina, Guzel Akhiyarova, Alla Korobova, Ekaterina I. Finkina, Dmitry Veselov, Tatiana V. Ovchinnikova, Guzel Kudoyarova

**Affiliations:** 1Ufa Institute of Biology, Ufa Federal Research Centre, the Russian Academy of Sciences, Prospekt Oktyabrya, 69, 450054 Ufa, Russia; vafinagh@mail.ru (G.V.); akhiyarova@rambler.ru (G.A.); muksin@mail.ru (A.K.); veselov@anrb.ru (D.V.); 2M.M. Shemyakin & Yu.A. Ovchinnikov Institute of Bioorganic Chemistry, Russian Academy of Sciences, 117997 Moscow, Russia; finkina@mail.ru (E.I.F.); ovch@ibch.ru (T.V.O.)

**Keywords:** jasmonates, lipid transfer proteins, long-distance transport, xylem sap, immunolocalization, immunoblotting, *Pisum sativum* L.

## Abstract

The adaption of plants to stressful environments depends on long-distance responses in plant organs, which themselves are remote from sites of perception of external stimuli. Jasmonic acid (JA) and its derivatives are known to be involved in plants’ adaptation to salinity. However, to our knowledge, the transport of JAs from roots to shoots has not been studied in relation to the responses of shoots to root salt treatment. We detected a salt-induced increase in the content of JAs in the roots, xylem sap, and leaves of pea plants related to changes in transpiration. Similarities between the localization of JA and lipid transfer proteins (LTPs) around vascular tissues were detected with immunohistochemistry, while immunoblotting revealed the presence of LTPs in the xylem sap of pea plants and its increase with salinity. Furthermore, we compared the effects of exogenous MeJA and salt treatment on the accumulation of JAs in leaves and their impact on transpiration. Our results indicate that salt-induced changes in JA concentrations in roots and xylem sap are the source of accumulation of these hormones in leaves leading to associated changes in transpiration. Furthermore, they suggest the possible involvement of LTPs in the loading/unloading of JAs into/from the xylem and its xylem transport.

## 1. Introduction

Jasmonic acid (JA) and its derivatives, called jasmonates (JAs) (jasmonic acid–isoleucine conjugate (JA-Ile), methyl jasmonate (MeJA), and others), are formed by the oxygenation of polyunsaturated fatty acids located in membranes [1]. JAs are involved in the regulation of plant development and responses to environmental stresses [2]. The successful adaptation of plants to stressful environments depends on signaling from the roots to shoots, which mediates a systemic response at the level of the entire plant [3]. Plant roots sense changes in the soil environment and may use hormones to provide shoots with an early warning of deteriorating soil conditions in ways that increase stress resistance [4]. The long-distance transport of jasmonic acid (JA) and its derivatives has been addressed mostly in regard to wound-induced systemic response ([5] and references therein). It has been shown that wounding induces the local accumulation of jasmonates in damaged leaves, as well as in distal, intact leaves, thereby triggering protective responses in both of them [6]. The results suggested the transmission of a mobile wound signal attributed to jasmonates. Although a more complex mechanism has been discovered that involves an electrical signal stimulating the production of JA in distal leaves [7,8], the translocation of jasmonates through the phloem, regulated by their transporters, has been demonstrated in Arabidopsis [9].

It has been discovered that JA and its derivatives are involved in responses to both biotic and abiotic stresses [6]. Thus, it has been shown that JAs play an important role in the response of plants to salt stress [10,11], for example, the salinity-induced expression of genes involved in JA biosynthesis [12] and increased JA content in leaves [13] and roots [14]. Furthermore, the level of JA accumulation was higher in salt-tolerant crops compared to sensitive varieties [15], while the use of JA significantly reduced salt-induced damage [16,17]. The adaption to salinity depends on responses in plant organs, which are remote from the site of perception of salt stimuli. For example, fast stomatal closure in salt-stressed barley seedlings maintained leaf hydration and the extension growth of leaf cells under the condition of a lowered water supply from the roots [18]. This was a response to the osmotic component of salinity (a decrease in the water potential of solution outside the plants due to addition of NaCl) before toxic sodium ions could accumulate inside the plants [19]. In a different study, reduced stomatal conductance under drought conditions depended on the delivery of abscisic acid (ABA) from the roots [4]. JA is also known to close stomata [20]. The concentration of JAs in xylem sap increased under drought conditions, and xylem-born JAs consistently decreased stomatal conductance in wild-type tomato shoots [21]. However, to our knowledge, the involvement of the long-distance transport of JAs from the roots to shoots has not been studied in relation to the control of stomatal closure in plants exposed to salt stress.

Another important aspect of JA transport over long distances is the possible participation of lipid transfer proteins (LTPs) in this process. Since JA is considered lipophilic [22], it is not surprising that LTPs were found to bind these hormones [23]. LTPs possess a hydrophobic cavity, inside which the ligand-binding site is located [24]. These proteins are able to reversibly bind various hydrophobic molecules, including plant hormones such as JA [25]. The ability of LTPs to bind plant JA was demonstrated using a fluorescence binding assay [26]. It is assumed that proteins of this type provide the necessary solubility of hydrophobic substances in hydrophilic spaces and enable their movement throughout the plants between membranes [25].

An immunohistochemical study of the presence of LTPs and ABA in the root cells of pea plants using specific antibodies revealed a similarity in their localization in the phloem cell walls and increased abundance of both under salt stress [27]. The results were interpreted as indicating the possible participation of LTPs in the unloading of ABA from the phloem. It was of interest to apply this approach to compare the distribution of JAs and LTPs in the cells of salt-stressed plants.

Root-predominant LTP was found in the xylem sap of soybean [28], which (in combination with the ability of LTP to bind JAs) suggests the possible involvement of this protein in the transport of lipophilic JAs through the hydrophilic environment of xylem sap. However, the presence of LTPs in the xylem sap of pea plants and their relationship with the level of JA in this compartment have not been studied under salinity conditions.

This report presents the results of a study of salt-induced changes in the content of JAs in roots and xylem sap, as well as of the effects of salt treatment on JA concentration in the leaves of pea plants related to the changes in transpiration. Although effects resulting from the accumulation of toxic ions in pea plants during long-term salinity conditions have been extensively studied [29], less attention has been paid to the rapid stomatal responses in pea plants exposed to NaCl. In addition, JA abundance and distribution between root and leaf cells were compared with the distribution of LTPs, using the immunohistochemical technique with specific antibodies in againt jasmonic acid and LTPs. We looked for similarities in their localization around vascular tissues involved in the transport process. Furthermore, we compared the effects of exogenous MeJA and salt treatment on the accumulation of JAs in leaves and their impact on transpiration. The overall purpose of this study was to verify whether salt-induced changes in JA concentrations in roots and xylem sap could explain the accumulation of these hormones in leaves and associated changes in transpiration, and to test a hypothesis that LTPs are involved in JA long-distance transport.

## 2. Results

At the first stage of the experiments, it was important to find out whether salt treatment affects the content of jasmonates at the site of perception of external influence (increasing salt concentration). The content of jasmonates in the roots of pea plants increased 1.5 h after the start of exposure to salt stress (Figure 1). Although the JA level in salt-treated plants decreased 4.5 h after the start of treatment, it remained higher than in the control. Fluorescence consistent with jasmonate abundance was most noticeable around the vascular and parenchyma cells of xylem.

Western blotting with the help of specific anti-LTP antibodies revealed the presence of these proteins in the xylem sap of pea plants, and an increase in their abundance under salinity (Figure 2).

A comparison of the abundance and distribution of JAs and LTPs between root cells revealed their similarity (Figure 1 and Figure 3). As with JA, fluorescence corresponding to the LTP content was very low in cross-sections of control roots, making cell boundaries barely visible. To reveal cells on the sections, we superimposed images obtained in the transmission mode on fluorescence images. Salt treatment increased LTP levels, resulting in clearly distinguishable xylem vascular and parenchyma cells. 

Next, we assessed the concentration of JAs in xylem sap and the rate of their release from the roots. JA concentrations were higher in the NaCl-treated than control plants when it was measured 1.5 h after the onset of exposure to salt stress (Table 1). The difference between the control and treated plants in the rate of release of JAs from the roots was less than in their concentration (7-times higher under salinity), which was due to the salt-induced decrease in the rate of xylem sap flow from the roots. Still, the delivery of hormones from the roots to shoots was increased by exposure to salt stress. The effect of salinity on both the concentration of JAs and the rate of their release from the roots decreased over time, but both indicators were still higher in the NaCl-treated plants than in control plants 4.5 h after the start of exposure to salt stress.

Then, it was necessary to detect any changes in transpiration as an indicator of shoot responses to salinity. 

Table 2 shows that salt treatment reduced the transpiration rate, and the extent of decline became greater with time: after 4.5 h, the reduction in transpiration was 1.5-times greater than 1.5 h after the start of exposure to salinity. In an attempt to link salt-induced changes in transpiration to hormonal responses, we studied the abundance of JAs on the leaf sections treated with specific antibodies. A total of 1.5 h after the onset of salt stress, no effect of salinity on the fluorescence level was detected. Fluorescence levels were low in both the control and salt-treated leaves. By this time, LTP-specific antibodies showed no salt-induced changes in LTP content in the leaves. Fluorescence corresponding to JA remained low in the leaves of the control plants 4.5 h after the start of the salt treatment (Figure 4a,b). Meanwhile, the JA level increased in the leaves of salt-treated plants (Figure 4d,e), although the LTP content did not change (Figure 4c,f). The immunohistochemical method revealed the presence of LTPs around the cells of leaf vascular bundles.

Since transpiration was reduced by salinity 1.5 h after the start of salt treatment, when the level of JAs in the leaves did not change, we looked for a factor that could cause a decrease in transpiration. Since ABA could play such a role, changes in its level and distribution in leaves were studied using immunohistochemistry.

Figure 5 shows the increased fluorescence of stomatal guard cells 1.5 h after the start of salt treatment, when antibodies against ABA were used. The increased extent of transpiration decline detected 4.5 h after the start of the salt treatment can be attributed to the additive effect of the accumulation of JAs in the leaves of salt-treated plants. To confirm the ability of JAs to reduce the transpiration of pea plants under these experimental conditions, we studied the effects of exogenous MeJA on transpiration and JA accumulation in pea leaves. Transpiration measurements showed its decrease at 2 h, when spraying the leaves with MeJA.

Figure 6 shows that this transpiration response was accompanied by the accumulation of JA in the leaves of treated plants, thereby confirming the ability of JA to decrease the transpiration of pea plants.

## 3. Discussion

In our previous article, using the immunohistochemical method, we showed the local effect of salt stress, which consists of the accumulation of JAs in the tips of pea roots resulting in the inhibition of root growth [14]. Root JA synthesis and perception controlled the transport of *de novo* synthesized nicotine to the leaves of wounded tobacco plants; however, these systemic events were not related to JA export from the roots to shoots [31]. The possibility of JA action as a long-distance signal transported from one plant organ to another has been most frequently discussed in the case of plant responses to wounding [5,9]. However, even in this case, the role of JAs as long-distance signals remains a matter of debate [7]. The study of JAs in xylem and their involvement in plant responses to abiotic stresses are very rare [21]. Currently, in search of evidence of the long-distance transport of JAs from roots to shoots, we studied the distribution and abundance of JAs in parts of roots, where the xylem was formed. As evidence in favor of the transport of JAs through the xylem, we found increased levels of these hormones around the root vascular bundle cells of pea plants exposed to salt stress. Our data are consistent with the report showing the expression of genes encoding dioxygenases (*DOXs*) and lipoxygenases (*LOXs*) involved in the synthesis of jasmoinc acids in the roots of transgenic lines containing promoter *GUS* constructs [32]. GUS activity in seedlings containing the *DOX* reporter gene was detected at the base of the roots, and *LOX* reporter gene activity was observed in the central cylinder of the root (Figure 3 of the cited article [32]). AtLOX6 was shown to be required for JA accumulation in roots upon abiotic and biotic stress [33], including salinity [34]. The similarity in the localizations of JAs and LTPs near xylem vessels, as well as the salt-induced increase in the abundance of both of them, can be explained by the participation of LTPs in the increased loading of JAs into the xylem of pea plants exposed to salt stress. Since LTPs are able to reversibly bind JAs [26], thereby ensuring the solubility of these hydrophobic substances in hydrophilic spaces, it can be assumed that LTPs facilitate the transport of JAs from living cells, where they are synthesized, to xylem vascular cells. The mechanisms by which LTPs influence the diffusion of hormones may involve not only the binding of JAs by LTPs, but also the induction of loosening the cell wall [35]. It was hypothesized by these authors that tobacco LTPs associate with hydrophobic wall compounds, causing the nonhydrolytic disruption of the cell wall. This ability of LTPs to influence cell wall structure may be important for facilitating the apoplastic transport of JAs. The involvement of JAs in systemic root-to-shoot signaling was supported by the data showing an increased concentration of JA in xylem sap and its greater release from the roots of salt-stressed pea plants. These data are consistent with the results of experiments conducted by De Ollas and others [21], who found an increased concentration of JA in the xylem sap of tomato plants under drought conditions. The presence of LTPs detected in the xylem sap of pea plants in the present experiments is an accordance with the results of the study of the xylem sap proteome of *Glycine max* [36]. The abundance of pea LTPs increased under salinity, mimicking salt-induced changes in JAs and supporting the possible involvement of LTPs in the transport of lipophilic JAs as well as other hydrophobic substances through the hydrophilic environment of xylem sap. The difference between LTP content in the xylem sap of the control and NaCl-treated plants was not great, and its biological significance should be confirmed by further experiments. In the present work, it was important to demonstrate the presence of LTPs in the xylem sap, which was performed for the first time for this type of LTP.

An increase in JA abundance in roots and their xylem concentration was followed, albeit with a delay, by the accumulation of JA in the leaves. Unlike the roots, the leaves of pea plants exposed to salt stress did not show an increase in LTP content. However, LTPs were present in vascular bundles and thus could participate in the unloading of JAs from the xylem. 

JAs are able to close stomata, and the accumulation of these hormones may be important for reducing transpiration [20], which maintains leaf hydration under the condition of a lowered water supply from the roots under salinity. However, jasmonic acid was shown to play a minor role in stomatal regulation in tomato plants [36]. So, it was important to find out if JA influences the transpiration of pea plants. Spraying pea leaves with a solution of MetJA increased JA abundance in the leaves and decreased transpiration. The increase in JA levels detected by anti-jasmonic acid antibodies in MeJA-treated leaves requires some comments. MeJA cannot be conjugated to proteins by the fixation method used in the present study because MeJA lacks the carboxyl group required for carbodiimide fixation. However, MeJA has been shown to be rapidly metabolized into jasmonic acid (JA) and jasmonoyl isoleucine (JI) [37], which can be conjugated to proteins by carbodiimide via their carboxyl groups, a prerequisite for their successful immunolocalization. The conversion of MeJA to JA and JI not only explains the increased levels of JAs detected by immunohistochemistry, but also the activation of the jasmonate signaling pathway [37], which leads to decreased transpiration. Thus, experiments with exogenous MeJA confirmed that the accumulation of JA in pea leaves can reduce the transpiration of salt-stressed plants. However, it remained unclear how transpiration was initially reduced in salt-stressed plants when increases in JA were not yet detected. In barley plants, rapid stomatal closure was associated with ABA accumulation in the leaves under salinity [18]. Considering these results, we studied the localization of ABA in salt-stressed pea plants, which increased the abundance of this hormone in stomatal guard cells under salinity. Thus, both ABA and JA obviously caused decreased transpiration in the salt-stressed pea plants. These results are consistent with the data showing that the foliar accumulation of both ABA and JA are required for maximum stomatal sensitivity to a water deficit in the soil.

## 4. Materials and Methods

### 4.1. Plant Growth and the Collection of Plant Material

Seeds of pea plants (*Pisum sativum* L.), variety “Sugar 2”, from the agricultural company “Aelita”, were disinfected before sowing with a mixture of 96% ethanol and 3% hydrogen peroxide (in a ratio of 3:1) for 10 min (the final concentration of ethanol was 72% and peroxide was 0.75%), followed by repeated washing with running and distilled water, in which they were left overnight to swell under continuously aerated conditions. The next day, the seeds were placed on a tray between layers of damp filter paper and covered with glass, leaving room for air to enter. Two days later, the seedlings were planted on a small floating device with a diameter of 8 cm made of foamed polyethylene (each with 8 holes for installing the seedlings). Plants were grown in a hydroponic culture on a 10% Hoagland–Arnon nutrient solution in deep containers with a volume of 5 L for a 14 h photoperiod, with illumination at 400–500 µmol m-2s-1PAR (ZN-500 and DNAT400 lamps) and temperature of 24/18 °C (day/night), while supporting the aeration of the solution. The solutions were replaced every day, and thus the pH was maintained at about 6.0 ± 0.2. Salt stress was created by transferring some of the 8-day-old seedlings to a solution of 50 mM sodium chloride. The solution of 50 mM NaCl was chosen because one of the important tasks of this work was the collection of xylem sap, which was difficult to perform at higher NaCl concentrations. The osmotic efficiency of this concentration of NaCl is confirmed by its effect on plant transpiration. A total of 1.5 and 4.5 h after the start of salinity exposure, pieces of plant roots and leaves were fixed for the immunolocalization of jasmonic acid, LTP, and ABA. To measure transpiration, polyethylene foam devices with installed seedlings were placed in 100 mL containers to weigh water loss for 15 min. Xylem exudate was collected to determine the quantitative content of JA 1.5 and 4.5 h after the start of salinity exposure. To collect xylem sap, 8-day-old seedlings were immersed in water, and the root system in the basal zone of the main root was separated with a sharp razor blade at an angle of 45°. A silicone tube of a suitable diameter was carefully placed on the stump and left for 1 h for exudation. Roots of control plants were placed into the Hoagland–Arnon nutrient solution, and salt-stressed plants were in the nutrient solution with 50 mM NaCl. The flow of jasmonates from roots was calculated by multiplying their concentration by the volume of xylem sap collected from one plant per h.

### 4.2. SDS Electrophoresis

Analysis of proteins was carried out by SDS-PAGE electrophoresis using the Laemmli method [38] in the tris-tricine system optimized to separate proteins with a molecular weight of less than 10 kDa [39]. Previously, the xylem sap was dialyzed (MWCO 6–8 kDa, Repligen) against a 100-fold volume of 50 mM ammonium acetate (pH 5.3) overnight at 4 °C. The amount of protein in the samples was determined by the Bradford method [30]. Xylem proteins were reduced by incubating them in a buffer containing β-mercaptoethanol (Sigma, St. Louis, MO, USA). Electrophoresis was carried out at constant currents of 15 mA (for concentration) and 25 mA (for separation) for 1–1.5 h.

### 4.3. Western Blotting

Wet transfer of proteins was carried out onto a nitrocellulose membrane (Bio-Rad, Feldkirchen, Germany) in a 25 mM sodium bicarbonate buffer (pH 9.0) containing 20% methanol and 0.1% SDS using the Mini Trans-Blot Electrophoretic Transfer Cell (Bio-Rad, Herts, UK) for 50 min at 15 °C and a constant current of 400 mA. Polyclonal rabbit antibodies against lentil Lc-LTP2 (anti-LTP, 1:100) cross-reacting with lipid transfer proteins from peas (Ps-LTP) were obtained and characterized as described [40]. Horseradish peroxidase-conjugated goat anti-rabbit IgG antibodies (1:10,000) (Sigma, St. Louis, MO, USA) were used. The nitrocellulose membrane was treated with a 3,3′,5,5′-tetramethylbenzidine liquid substrate system (Sigma, St. Louis, MO, USA). The total intensity staining of the protein bands was quantified using ImageJ 1.54 g Wayne Rasband and contributors from the National Institutes of Health, USA (http://imagej.org, accessed on 18 October 2023).

### 4.4. Methyl Jasmonate Treatment

Methyl jasmonate (Sigma, St. Louis, MO, USA) was dissolved in 80% ethanol and added to a 0.05% solution of Tween 20 in distilled water at a final hormone concentration of 10 mM. The ethanol concentration in the final solution was insignificant and amounted to less than 0.01%. The selected concentration of methyl jasmonate for the exogenous treatment of pea plants was selected in preliminary experiments as helping to suppress plant transpiration. Shoots of pea plants in the stage of 2 true leaves were sprayed with a solution of methyl jasmonate, avoiding its entry into the nutrient solution. Shoots of control plants were sprayed with a solution of Tween 20 and ethanol at the same concentration. The selection of leaves for the localization of jasmonic acid by immunohistochemistry was carried out 60 min after the start of treatment of plants with methyl jasmonate.

### 4.5. Immunoassay of Jasmonates

After collecting and weighing the xylem sap, ethanol was added to a final concentration of 80%, and the solution was incubated overnight at 4 °C. After the evaporation of ethanol, the JAs were extracted twice with dimethyl ether from the aqueous residue acidified to pH 2.5 at a ratio of the organic and aqueous phases of 1:3 [41]. An amount of 1% of the NaHCO3 solution was added to the combined organic fraction (the organic/aqueous phase ratio was 3:1), shaken, and the ether was discarded. After the acidification of the aqueous solution containing hormones, ether re-extraction was carried out by reducing the volume of the extractant (3:1). Reducing the amount of extractant at each stage of extraction and re-extraction increases the selectivity of hormone recovery [42]. After the evaporation of the diethyl ether, the hormones were dissolved in 80% ethanol and aliquots were taken for quantification by ELISA using specific antibodies to jasmonic acid (1:6000) (Agrisera, Vannas, Sweden) as described [43]. Since antibodies to jasmonic acid were obtained by Agrisera using BSA-conjugated jasmonic acid as an immunogen, these antibodies recognize both free JA and its conjugates (e.g., methyl jasmonate). However, the modified extraction scheme frees the extract from methyl jasmonate at the stage of extraction from the dimethyl ether into the NaHCO_3_ solution and allows the preferential recovery of free JA.

### 4.6. Immunohistochemistry

Immunolocalization of JA, ABA, and LTP was carried out in the leaves and basal part of roots of pea plants as described [27]. Five-millimeter-long tissue pieces were collected and fixed in phosphate buffered saline (PBS) (pH 7.2) containing 4% N-(3-dimethylaminopropyl)-N’-ethylcarbodiimide hydrochloride (Merck, Darmstadt, Germany) for 6 h, and then in 4% formaldehyde (Riedel de Haen, Seelze, Germany) and 0.1% glutaraldehyde (Sigma, Steinheim, Germany) at 4 °C overnight. To improve the penetration of reagents into plant tissues, a vacuum was applied for the first 30 min. Then, the plant pieces were transferred to fresh fixative and left overnight at 4 °C. After dehydration in an ascending ethanol series, plant tissues were embedded in JB4 resin (Electron Microscopy Sciences, Hatfield, PA, USA). Histological sections with a thickness of 1.5 μm were obtained with an HM 325 MICROM microtome (Laborgerate, Walldorf, Germany) and placed on glass slides. Samples were pre-treated with phosphate buffered saline containing 0.2% gelatin and 0.05% Tween-20 (PGT) for 30 min to reduce non-specific binding, and then incubated overnight at 4 °C with antibodies raised against either JA (Agrisera, Vannas, Sweden), LTP [40], or ABA [27]. After washing three times with 0.05% Tween-20 in PBS for 10 min, sections were incubated with secondary antibodies raised against rabbit immunoglobulins conjugated to Alexa Fluor 555 and 488 (Invitrogen, Rockford, IL, USA) (1:500 in PGT) for 3 h at 37 °C in darkness. Alexa Fluor 488 was used in the case of leaf sections to avoid the fluorescence of chlorophyll. Samples were washed five times for 10 min with PBS and rinsed with MilliQ water. Then, the sections were coated with a mixture of glycerin and gelatin (0.5 g of gelatin and 3.5 mL of glycerin in 3 mL of water), immediately covered with slips, and viewed with a confocal laser scanning microscope using an FV3000 FluoView (FV31-HSD) (Olympus, Tokyo, Japan) and laser excitation of 488 or 561 nm. Fluorescence emission was detected at 520 nm (Alexa 488) and 568 nm (Alexa 555) in the integration frame mode for imaging with a count of 2. To facilitate the discrimination between LTPs and JAs, we assigned them to different pseudocolors indicated in the figure legends, when they are presented in the same figure. Thus, in Figure 4, the cyan pseudocolor is assigned to JA and the yellow pseudocolor is assigned to LTP.

### 4.7. Statistics

The data were processed using Statistica version 10 software (Statsoft, Moscow, Russia). In the tables, the data are presented as mean ± standard error (s.e.). The significance of differences was assessed by ANOVA, followed by Duncan’s test (*p* < 0.05). The number of replications is provided in the table legends.

## 5. Conclusions

The obtained results confirm that JA produced in roots acts as a long-distance signal transported through the xylem, and its involvement in the control of transpiration under salinity and LTPs is likely to participate in its loading into xylem and transport through xylem. JAs are likely to act in concert with ABA, which is in accordance with the literature data.

## Figures and Tables

**Figure 1 ijms-25-07486-f001:**
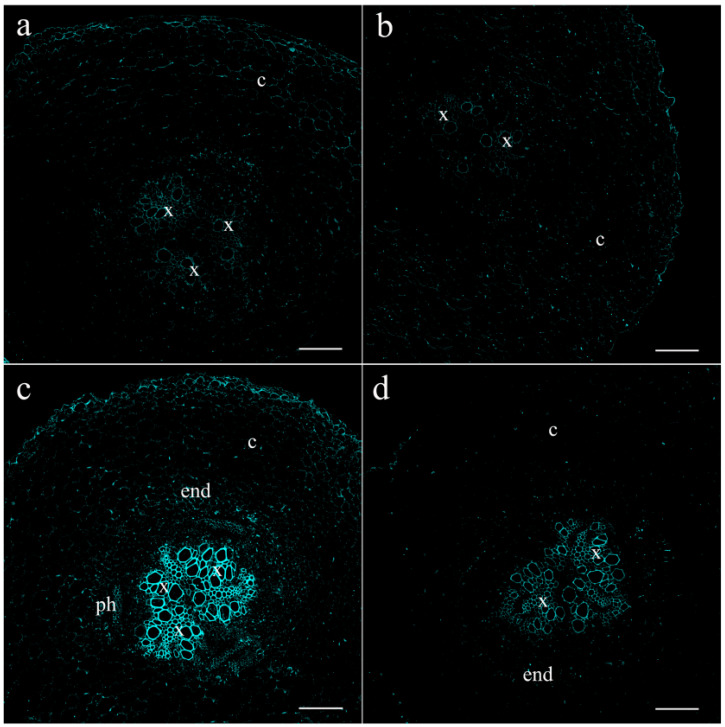
Effects of salt treatment on JA localization in the pea root basal part 1.5 (**a**,**c**) and 4.5 (**b**,**d**) hours after the onset of salt exposure. Root section of control pea plants untreated with NaCl (**a**,**b**) and plants treated with NaCl (**c**,**d**). The scale bar is 150 µm. end—endodermis; x—xylem; ph—phloem; c—cortex.

**Figure 2 ijms-25-07486-f002:**
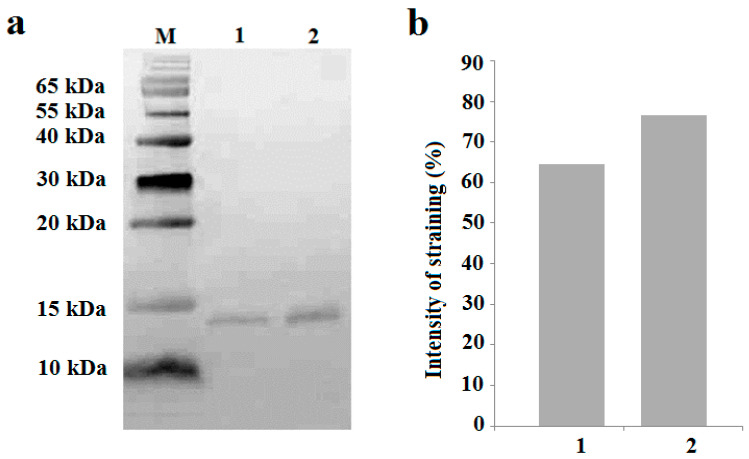
Immunoblotting with polyclonal rabbit anti-LTP antibodies of xylem sap of pea plants untreated with NaCl (1) and 1.5 h after the onset of salt exposure (2), total protein 20 μg; M- molecular weight marker (**a**). Relative staining, arbitrary units, maximal staining taken as 100%, and minimal as 0% (**b**). The amount of protein in the samples was determined by the Bradford method [30], and equal amounts were loaded for electrophoresis.

**Figure 3 ijms-25-07486-f003:**
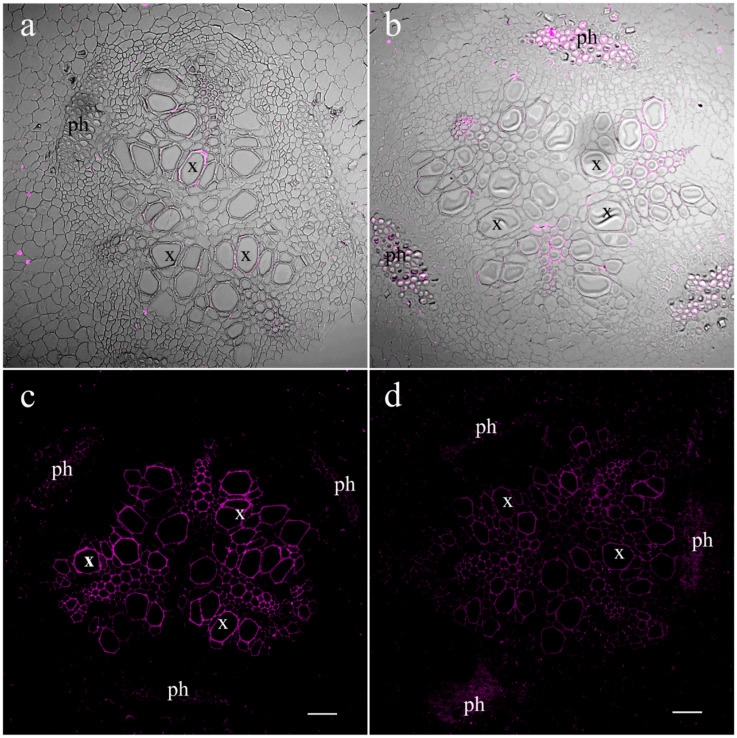
Effects of salt treatment on the level and localization of lipid transfer proteins (LTPs) in the central cylinder of the root basal part 1.5 (**a**,**c**) and 4.5 (**b**,**d**) hours after the onset of salt exposure. Root section of control pea plants untreated with NaCl (**a**,**b**) and plants treated with NaCl (**c**,**d**). In (**a**,**b**), transmission mode images are superimposed on fluorescence images. The scale bar is 50 µm. x—xylem; ph—phloem.

**Figure 4 ijms-25-07486-f004:**
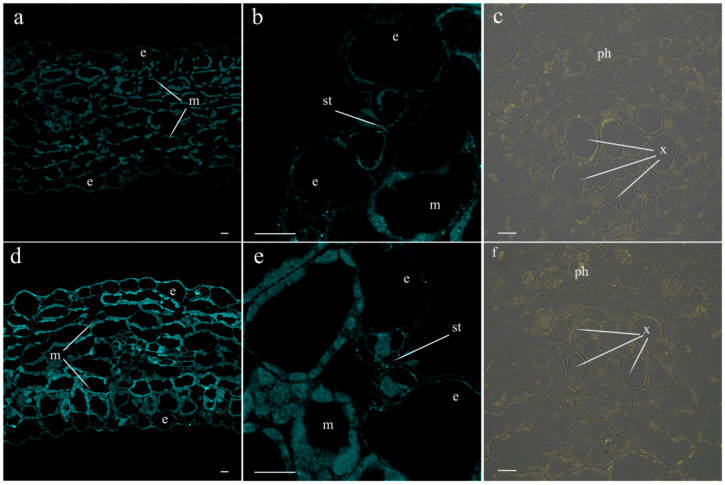
Effects of salt treatment on the level and localization of JAs (**a**,**b**,**d**,**e**) and LTPs (**c**,**f**) in the leaves 4.5 h after the onset of salt exposure. Leaf section of control pea plants untreated with NaCl (**a**–**c**) and plants treated with NaCl (**d**–**f**). (**a**,**d**)—leaf tissues, (**b**,**e**)—stomata, and (**c**,**f**)—leaf vascular bundles. The scale bar is 10 µm. e—epidermis, m—mesophyll, st—stomata, x—xylem, and ph—phloem. To facilitate discrimination between JAs and LTPs detected by the same label (Alexa 488), we assigned them to different pseudocolors: the cyan pseudocolor is assigned to JAs and the yellow pseudocolor is assigned to LTPs.

**Figure 5 ijms-25-07486-f005:**
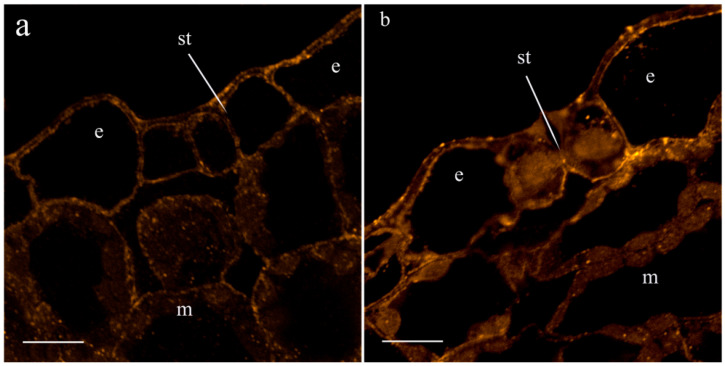
Effects of salt treatment on the level and localization of ABA in the stomata of pea plants 1.5 h after the onset of salt exposure. Leaf section of control pea plants untreated with NaCl (**a**) and plants treated with NaCl (**b**). The scale bar is 10 µm. e—epidermis, m—mesophyll, and st—stomata.

**Figure 6 ijms-25-07486-f006:**
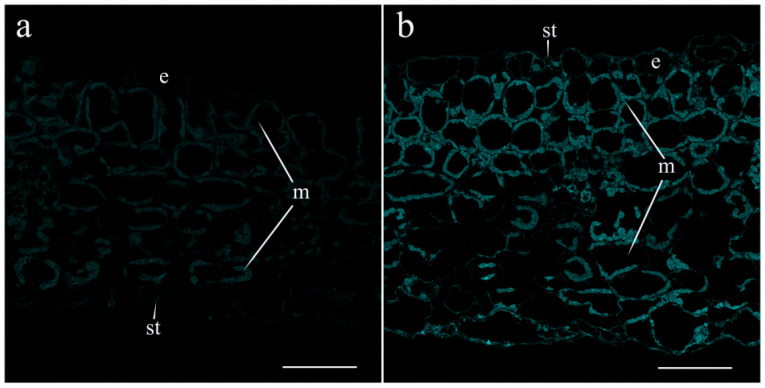
Effect of exogenous MeJA (10 mM) on the level and localization of JA in the leaves of pea plants 1 h after the onset of shoot treatment. Leaf section of control pea plants untreated with meJA (**a**) and plants treated with meJA (**b**). The scale bar is 50 µm. e—epidermis, m—mesophyll, and st—stomata.

**Table 1 ijms-25-07486-t001:** Effect of salt stress on JA concentration in xylem sap (ng mL^−1^) and rate of JA flow from the roots (ng plant^−1^ h^−1^) of pea plants. Mean ± standard error (n = 10) are presented. Means significantly different from the control (without NaCl treatment) are marked with asterisk (*t*-test, *p* ≤ 0.05).

Treatments	Time after the Start of Treatment, h			
	1.5		4.5	
	Concentration	Flow Rate	Concentration	Flow Rate
Control	12 ± 4	0.97 ± 0.1	36 ± 5	0.41 ± 0.1
NaCl treatment	206 ± 19 *	6.9 ± 0.7 *	224 ± 30 *	0.62 ± 0.1 *

**Table 2 ijms-25-07486-t002:** Effect of salt stress on transpiration (mg h^−1^ plant^−1^). Mean ± standard error (n = 50) are presented. Means that do not differ statistically from each other are marked with the same letters. (*p* ≤ 0.05, Duncan’s test).

Treatments	Time after the Start of Treatment, h	
	1.5	4.5
Control	119 ± 3 ^c^	118 ± 3 ^c^
NaCl treatment	99 ± 1 ^b^	90 ± 1 ^a^

## Data Availability

Data are contained in the article.

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
