# Peer review of "The Long-Distance Transport of Jasmonates in Salt-Treated Pea Plants and Involvement of Lipid Transfer Proteins in the Process"

_ijms, 2024, doi:10.3390/ijms25137486_

Round 1
Reviewer 1 Report
Comments and Suggestions for Authors
The title "Long-Distance Transport of Jasmonates in Salt-Treated Pea Plants and Involvement of Lipid Transfer Proteins in the Process" is intriguing and aligns well with the journal's scope. The authors have made a commendable effort, but a few minor revisions are necessary.
- Please clarify why you chose 50 mM sodium chloride as the standard level for salinity in your experiment.
- Correct the title and units in lines 146 and 147.
- The statistics in lines 360-366 need to be rewritten for clarity and accuracy.
Comments on the Quality of English Language
Additionally, minor editing of the English language is required throughout the manuscript.
Author Response
- The title "Long-Distance Transport of Jasmonates in Salt-Treated Pea Plants and Involvement of Lipid Transfer Proteins in the Process" is intriguing and aligns well with the journal's scope. The authors have made a commendable effort, but a few minor revisions are necessary. Response: We are grateful to the respected reviewer for king words about our article.
- Please clarify why you chose 50 mM sodium chloride as the standard level for salinity in your experiment. Response: A solution of 50 mM NaCl was chosen because one of the important tasks in this work was collection of xylem sap, which was difficult to do at higher NaCl concentrations. The osmotic efficiency of this concentration of NaCl is confirmed by its effect on plant transpiration. This explanation is introduced into the M & M section
- Correct the title and units in lines 146 and 147.Response: The title was modified “Effect of salt stress on transpiration (mg h-1 plant -1). Means ± standard error (n=50) are presented. Means that do not differ from each other are marked with the same letters. (p ≤ 0.05, Duncan test).” We also decided to provide some explanations. In the present experiments plants did lose the indicated mg of water per plant during 1 h. So this unit (mg h-1 plant -1 ) was indicated. The present formulation concerning statistics was used, since we compared all means with each other and not only with the control
- The statistics in lines 360-366 need to be rewritten for clarity and accuracy. Response: We are sorry for providing two variants of statistics description. The first of them was deleted.
Reviewer 2 Report
Comments and Suggestions for Authors
Effect of salt stress on JA accumulation and transport have been studied in the current paper. Authors applied new immunolocalization methods in situ.
The topic is interesting and some results are significantly new. However, some points require clarifications and corrections.
The first question which required explanations is timing : 1,5 h and 4,5 h. How fast sodium uptake by pea root? Where Na is localize : cytoplasm or vacuole? This need to be explained and discussed. In previous publications authors use few days after Na uptake and usually sodium effect is more prolonged. See example.
https://www.ncbi.nlm.nih.gov/pmc/articles/PMC9865415/.
The secong point is antibody specificity: which form of JA they recognise?
JA or MeJA? Conjugated or free?
JA have a lot of form with different activity, indeed..
https://link.springer.com/chapter/10.1007/978-981-99-5736-1_21
Lines 50 – 51: may be it is logical to first mention biosynthesis, and thereafter accumulations??
Figure 2: please, move a and b to left size. Loading control are required.
Figure 3: panel A and B – I can not see nothing. I would suggest to co-stain with SR2200 or with BR28 (calcofluor white). It is really improve understanding of the images.
Line 265: maybe it is better to mention final concentrations of C2H5OH andH2O2 ?
Lines 274- 275: I am not sure everyday re-fresh a medium is an optimal way. Plants adapted medium for better growth.
Line 302: what is primary Ab source? Company? Own?
Lines 317 – 318: how did you wash from surface-binding MeJ?
Line 336: „4% paraformaldehyde“?? It should be formaldehyde, para mean a powder. Once you dissolve paraformaldehyde, it became formaldehyde.
Line 338: „fixation was carried out under vacuum for the first 30 minutes.“ ??? Under vacuum fixation did not occurred, fixation start after removing vacuum. Vacuum require only for very short time.
Lines 357- 359: please, re-write more clearly.
Lines 360 and 363: doubling “statistics”?
Comments on the Quality of English LanguageMinor polishing
Author Response
Comment 1:
Effect of salt stress on JA accumulation and transport have been studied in the current paper. Authors applied new immunolocalization methods in situ. The topic is interesting and some results are significantly new. However, some points require clarifications and corrections.
Response 1: We are most grateful to the respected reviewer for attentive analysis of our article and valuable comments, which helped us to improve the MS
Comment 2. The first question which required explanations is timing : 1,5 h and 4,5 h. How fast sodium uptake by pea root? Where Na is localize : cytoplasm or vacuole? This need to be explained and discussed. In previous publications authors use few days after Na uptake and usually sodium effect is more prolonged. See example. https://www.ncbi.nlm.nih.gov/pmc/articles/PMC9865415/.
Response 2: This question showed us that while justifying the design of our experimental approach we did not pay enough attention to the two components of salinity action on plants. Those are (1) osmotic component, which acts immediately after the start of salt addition and is due to the presence of osmotically active NaCl outside the plants and the second is its toxic component (2), which affects plants later on and results from accumulation of sodium inside the plants. In our article we concentrated on the first of them, since we were interested in the involvement of jasmonates in fast stomatal responses. Such fast stomatal responses to salinity have been shown by us in barley plants (Fricke et al., 2004). To clarify this in accordance with the remark of respected reviewer, we added to the Introduction that “This rapid stomatal closure was a response to the osmotic component of salinity (a decrease in the water potential of solution outside the plants due to addition of NaCl), which occurs before toxic sodium ions accumulate inside the plants [Munns and Tester, 2008].” We also added that “Although effects resulting from accumulation of toxic ions in pea plants during prolonged salinity have been thoroughly studied [Popova et al.,2023], fast stomatal responses in NaCl stressed pea plants received less attention.”. Referance to Popova et al. was recommended by reviewer.
Comment 3. The secong point is antibody specificity: which form of JA they recognise? JA or MeJA? Conjugated or free? JA have a lot of form with different activity, indeed.https://link.springer.com/chapter/10.1007/978-981-99-5736-1_21
Response 3: We are sorry that we did not provide sufficient information concerning specificity of antibodies in the initial variant of the article. In accordance with the valuable comment of the referee we added to the M & M section that “ Since antibodies to jasmonic acid were obtained by Agrisera using BSA-conjugated jasmonic acid as immunogen, these antibodies recognize both free JA and its conjugates (e.g., methyl jasmonate). However, the modified extraction scheme frees the extract from methyl jasmonate at the stage of extraction from dimethyl ether into NaHCO3 solution and allows the preferential recovery of free JA.” Furthermore, this question is also addressed in Discussion, where we wrote that “MeJA cannot be conjugated to proteins by the fixation method used in the present study because MeJA lacks the carboxyl group required for carbodiimide fixation. However, MeJA has been shown to be rapidly metabolized into jasmonic acid (JA) and jasmonoyl isoleucine (JI) [38], which can be conjugated to proteins by carbodiimide via their carboxyl groups, a prerequisite for their successful immunolocalization.” Different forms of jasmonates are specified in the first sentence of Introduction, where it is said that “Jasmonic acid (JA) and its derivatives called jasmonates (JAs) (jasmonic acid-isoleucine conjugate (JA-Ile), methyl jasmonate (MeJA) and others) are formed by oxygenation of polyunsaturated fatty acids located in membranes [1].”
Comment 4. Lines 50 – 51: may be it is logical to first mention biosynthesis, and thereafter accumulations??
Response 4: In accordance with this comment, we modified the sentence: “salinity induced expression of the genes involved in JA biosynthesis [12] and increased JA content in leaves [13] and roots [14]”
Comment 5. Figure 2: please, move a and b to left size. Loading control are required.
Response 5: a and b were moved to the left. We added to the Figure legend that “The amount of protein in the samples was determined by the Bradford method [30] and equal amounts were loaded for electrophoresis”. We realize that the difference between LTP content in xylem sap of control and NaCl treated plants is not great and its biological significance should be confirmed by further experiments. In the present work it was important just to demonstrate the presence of LTP in the xylem sap, which was done for the first time for this type of LTP. To emphasize importance of identification of LTP in xylem sap, we added this information into the text.
Comment 6. Figure 3: panel A and B – I can not see nothing. I would suggest to co-stain with SR2200 or with BR28 (calcofluor white). It is really improve understanding of the images.
Response 6. We are sorry for this, but fluorescence was really low in the control, which was likely to be due to low abundance of LTP in this root region, when NaCl was not added to the plants. To reveal cells on the sections in the revised version we superimposed images obtained in transmission mode on fluorescence images. We hope that in this was we managed to fulfill recommendation of respected reviewer. In accordance, it is added to the figure legend that “In Figures a and b, transmission mode images are superimposed on fluorescence images.” We also tried to use RS2200 according to recommendation of respected reviewer. But its fluorescence was so bright that fluorescence corresponding to LTP, which was very low, could not be distinguished. So we have chosen transmission mode, which allowed to see fluorescence corresponding to LTP
Comment 7. Line 265: maybe it is better to mention final concentrations of C2H5OH andH2O2
Response 7. According to this recommendation we added that “the final concentration of ethanol was 72% and peroxide was 0.75”
Comment 8. Lines 274- 275: I am not sure everyday re-fresh a medium is an optimal way. Plants adapted medium for better growth.
Response 8. We need to draw attention of respected reviewer that in our experiments we used 10 % Hoagland-Arnon solution, since in previous experiments it was found that with this concentration nutrient solution contained optimal concentration of mineral nutrients to maintain plants growth. Still mineral nutrients can be quickly absorbed by plants from this diluted solution, and therefore we changed it for the fresh solution each day.
Comment 9. Line 302: what is primary Ab source? Company? Own?
Response 9. Antibodies against LTP were obtained by co-authors of the present article at the Institute of Institute of Bioorganic Chemistry. Purification of LTP, immunization scheme and cross- reactivity of obtained antibodies are described in the details in the article cited in the M & M section of our MS: “Ð olyclonal rabbit antibodies against lentil Lc-LTP2 (anti-LTP, 1:100) cross-reacting with lipid transfer proteins from pea (Ps-LTP) were obtained and characterized as described [38].” We separated this sentence from the description of the second antibodies to make this clearer.
Response 10. Leaves were washed with water before cutting sections for immunolocalization. We understand that this could be not sufficient to wash hydrophobic MeJ from the leaf surface. But as can be seen from Figure 6, no fluorescence corresponding to jasmonates was detected on the leaf surface. This was possibly due to washing of MeJ from the leaf surface during dehydration with ethanol, while washing out of jasmonic acid formed from MeJ inside the leaf was prevented by conjugation with carbodiimade prior to dehydration.
Comment 11. Line 336: „4% paraformaldehyde“?? It should be formaldehyde, para mean a powder. Once you dissolve paraformaldehyde, it became formaldehyde.
Response 11. Thanks for valuable explanation! “para” is deleted
Comment 12. Line 338: „fixation was carried out under vacuum for the first 30 minutes.“ ??? Under vacuum fixation did not occurred, fixation start after removing vacuum. Vacuum require only for very short time.
Response 12. In response to this comment, the sentence was modified “To improve penetration of reagents into plant tissues, vacuum was applied for the first 30 minutes.” We are grateful for recommendation to shorten vacuum exposure, but we cannot repeat all experiments once again. At least, we did not detect disruption of the cell structures with the present procedure.
Comment 13. Respected reviewer recommended minor revision of English of our article.
Response 13. According to the comment of reviewer, English of our article was kindly corrected by our colleague, whose first language is English.
Round 2
Reviewer 2 Report
Comments and Suggestions for Authors
Response 6:
Thanks! Maybe you can reduce exposition time to BR28? nAnyway, now it is better visible.
Response 8. We need to draw attention of respected reviewer that in our experiments we used 10 % Hoagland-Arnon solution, since in previous experiments it was found that with this concentration nutrient solution contained optimal concentration of mineral nutrients to maintain plants growth. Still mineral nutrients can be quickly absorbed by plants from this diluted solution, and therefore we changed it for the fresh solution each day.
Thank you for explanations. I think it will be better to not fully re-fresh, but re-fresh 50- 70%. By the way, have you controlled pH of the solution before and after? Have you add stabiliser for create complexones?
Comments 12:
The problem is that fixation starting ONLY after vaccuum gone, when fixer penétrate inside tissue. Under vaccuum fixer can not penétrate inside tissue. Plesae, consider for future.
Comments on the Quality of English LanguageProof-reading
Author Response
Сomment 1. Thanks! Maybe you can reduce exposition time to BR28? nAnyway, now it is better visible.
Response 1. Thanks for you advise. We shall reduce exposition time when we use BR28 next time. But we are glad that you found the figure better visible in the revised version.
Comment 2. Thank you for explanations. I think it will be better to not fully re-fresh, but re-fresh 50- 70%. By the way, have you controlled pH of the solution before and after? Have you add stabiliser for create complexones?
Response 2. Thanks for your advice to refresh only 50-70 % of the nutrient solution. We shall bare this in mind in the future. pH of the nutrient solution was about 6 and slightly increased after one day of growing plants on it before its refreshing (possibly due to the uptake of nitrates, which is known to occur in symport with hydrogen ions). But it remained in the range of optimal values for plant growing, which is between 5.5 and 6.5. According to the comment of respected reviewer we added to M & M section that “and thus its pH was maintained about 6.0±0.2”
Comments 3: The problem is that fixation starting ONLY after vaccuum gone, when fixer penétrate inside tissue. Under vaccuum fixer can not penétrate inside tissue. Please, consider for future.
Response 3. Thanks for one more valuable advice. We shall shorten vacuum exposure in the future and repeat it, if necessary.